# ResNet-AE for Radar Signal Anomaly Detection

**DOI:** 10.3390/s22166249

**Published:** 2022-08-19

**Authors:** Donghang Cheng, Youchen Fan, Shengliang Fang, Mengtao Wang, Han Liu

**Affiliations:** School of Space Information, Space Engineering University, Beijing 101416, China

**Keywords:** deep learning, anomaly detection, autoencoder, residual networks, LSTM

## Abstract

Radar signal anomaly detection is an effective method to detect potential threat targets. Given the low *Accuracy* of the traditional AE model and the complex network of GAN, an anomaly detection method based on ResNet-AE is proposed. In this method, CNN is used to extract features and learn the potential distribution law of data. LSTM is used to discover the time dependence of data. ResNet is used to alleviate the problem of gradient loss and improve the efficiency of the deep network. Firstly, the signal subsequence is extracted according to the pulse’s rising edge and falling edge. Then, the normal radar signal data are used for model training, and the mean square error distance is used to calculate the error between the reconstructed data and the original data. Finally, the adaptive threshold is used to determine the anomaly. Experimental results show that the recognition *Accuracy* of this method can reach more than 85%. Compared with AE, CNN-AE, LSTM-AE, LSTM-GAN, LSTM-based VAE-GAN, and other models, *Accuracy* is increased by more than 4%, and it is improved in *Precision*, *Recall*, *F*1-score, and AUC. Moreover, the model has a simple structure, strong stability, and certain universality. It has good performance under different SNRs.

## 1. Introduction

With the rapid development of information technology, the importance of the information battlefield has become increasingly prominent. The traditional land, sea, and air three-dimensional space situation has been unable to meet the needs of the modern battlefield environment, and the battlefield space has been expanded to electromagnetic space. As a symbol element of the information battlefield, the electromagnetic situation has attracted much attention since it was put forward. Under the condition of modern information technology, various information warfare platforms and electronic equipment have been put into the information war, enabling electronic equipment to obtain a large number of time sequence signal data in a short time. Through the abnormal detection of these time sequence signals, the time nodes with anomalies can be found as soon as possible, which is of great significance for analyzing the enemy situation, eliminating hidden dangers, and assisting decision making.

Anomaly detection is screening situations contrary to the distribution law of normal data from the data to be detected [1,2,3,4]. The traditional anomaly detection model [5,6,7] uses complex algorithms and equipment. It has poor real-time performance and cannot be popularized. Scholars at home and abroad have proposed many unsupervised learning methods to solve the problems. Standard methods include the AE-based method and the GAN-based method [8,9]. The former extracts the potential features of time-series signals by establishing neural networks, reconstructs the signals by features, and distinguishes whether the reconstructed signals are abnormal by evaluating the differences between the reconstructed signals and the original signals. The latter reconstructs the timing signal through the generator, and the discriminator judges whether it is an anomaly. The two continue to iterate and optimize to achieve the desired effect. In 2015, An et al. proposed an anomaly detection method using VAE to reconstruct probability [10], which is better than the methods based on an autoencoder and a principal component. Using the generation characteristics of the VAE, the data can be reconstructed, and the root cause of the anomaly can be analyzed. In 2016, O’Shea et al. proposed a periodic anomaly detector [11] that models and predicts IQ channel data. It uses an LSTM network model to predict IQ channel sampling data of the following four times by learning the past signal sampling values of 32 IQ channels. Then, it judges whether there is an anomaly based on the error value. However, this method depends on the periodic change law of electromagnetic signals, and the periodicity of electromagnetic signals often changes with time. Therefore, this method can only predict short-term anomalies, which has great limitations. In 2018, Xu et al. constructed the donut algorithm [12] based on the VAE, which trains the normal and abnormal data simultaneously, making the feature extraction more complete and providing a new idea for the VAE-based anomaly detection algorithm. In 2019, Chen et al. used the confrontational training method, Buzz [13], to detect anomalies in complex time series. This method not only reached a very high level in the public dataset but also gave a theoretical inference to transform the model into a Bayesian network, which enhanced the interpretability of the model. In 2020, Niu et al. proposed a VAE-GAN detection model [14]. The model jointly trains the encoder, generator, and discriminator, which can improve the fidelity of signal reconstruction, make the distinction between normal and anomalies more significant, and improve anomaly detection *Accuracy*. Lin et al. proposed a mixed anomaly detection method [15], which combines the representation learning ability of the VAE with the time modeling ability of the LSTM. The VAE structure aims to capture the structural rules of the time subsequence on the local window, while the LSTM structure models the changing trend of the long-term time series. Audibert et al. proposed a fast and stable unsupervised anomaly detection method, USAD, for multivariate time series [16]. This method uses the automatic encoder architecture to meet the conditions of unsupervised learning. The use of adversarial training enables rapid training and the isolation of anomalies. Experiments on five public and proprietary datasets verify its high robustness, training speed, and anomaly detection performance. Huang et al. proposed an unsupervised time-series anomaly detection method based on multimodal countermeasure learning [17], which converts the original time series into frequency-domain space, constructs a multimodal time series representation, uses a multimodal generation countermeasure network model, and realizes unsupervised joint learning of normal time-series information about time-domain and frequency-domain feature distribution for multimodal time series, The anomaly detection problem is transformed into the measurement problem of time-series reconstruction in time-frequency space. The anomaly of time series is measured from the time domain and frequency domain. Compared with the traditional single-mode method, this method has improved the AUC and AP by 12.50% and 21.59%, respectively, which provides a new direction for electromagnetic signal anomaly detection based on deep learning.

However, GAN networks tend to have complex structures and a high overhead of training and detection, while traditional AE networks have low *Accuracy* in anomaly detection of electromagnetic signals. Therefore, this paper proposes a ResNet-AE network model based on the AE network. This model uses the encoder and decoder with ResNet for feature mapping and data reconstruction. It can effectively alleviate the problem of gradient disappearance, improve the depth of the network that can be effectively trained, and improve the ability of network feature extraction and reconstruction. LSTM is used to acquire time-dependent features. Cluster analysis is used to process the anomaly detection results to obtain an adaptive decision threshold.

To summarize, the main contributions of our work are:An anomaly detection method based on ResNet-AE is proposed to detect radar signal time-series data.Our method is jointly ResNet and autoencoder, which takes good feature extraction and reconstruction capabilities.Anomaly score is an adaptive threshold obtained by clustering the reconstructed difference, which makes it more able to distinguish anomalies from normal data.

## 2. ResNet-AE Anomaly Detection Model

### 2.1. Dataset

The dataset selected for the experiment is the actual FMCW radar signal. The carrier frequency is about 100 MHz, and the sampling rate is 400 msps. The visual image of the signal is shown in Figure 1. Five groups of data are selected. Each group is the emitter signals of different radars, composed of 1000 pulse signals of time sequence. Each pulse signal is extracted into time sequence subsequence samples and divided into the training, verification, and test sets by setting the threshold of pulse rising and falling edge. Through down conversion and resampling, each subsequence sample contains 100 sampling points. The abnormal pulse signal is randomly added to the test set so that the abnormal ratio of the test set is 15%.

Before the experiment, it is necessary to carry out data preprocessing, normalize all values in the dataset to the [0, 1] interval with the maximum and minimum normalization, and arrange the intercepted signal subsequences in time order so that the temporal correlation can be preserved after being input into the network.

### 2.2. Model Construction and Training

In this paper, the ResNet-AE model is proposed for anomaly detection. The model takes the autoencoder as the main network framework, and the encoder and decoder are stacked by residual structure. Each residual module comprises CNN, pooling layer, LSTM, and ReLU activation function. The network structure is shown in Figure 2.

#### 2.2.1. Network Structure

The method has two stages: the model training stage and the model testing stage, as shown in Figure 2a. In the training stage, the training data were input into ResNet-AE, and the encoder was used to extract features to obtain the potential distribution rules of the data. Then, the original data were reconstructed by the decoder, and the network model was optimized by continuously reducing the error between the reconstructed data and the original data. After the training, the parameters in the network will not change, and then the testing data will be input into the network. The reconstructed data will be obtained through network calculation. The anomaly decision will be made based on the adaptive threshold and the error between the reconstructed and original data. In ResNet-AE, CNN can efficiently extract the characteristics of data and learn the potential distribution law of time-series signals, LSTM can learn the time dependence of data, and residual structure can effectively solve the vanishing gradient problem, which makes the deep network better learning and reconstruction ability.

#### 2.2.2. ResNet-AE

Based on the traditional AE network, the ResNet-AE network replaces the linear structure in the AE network with the ResNet structure. The encoder and decoder are stacked by ResNet modules. The ResNet-AE network consists of four encoders and three decoders. Each encoder contains two modules: one-dimensional convolution layer, LSTM layer, ReLU activation function, lower sampling layer, etc. The module structure of the encoder is shown in Figure 2b. The shape of each subsequence of the input network is 100 × 1, and four encoders extract the feature. The feature space extracted by each encoder is 64 × 1, 32 × 1, 16 × 1, and 8 × 1, respectively. The structure of the decoder is contrary to that of the encoder, as shown in Figure 2c, which is composed of an upper sampling layer, an LSTM layer, and two one-dimensional convolution layers and uses ReLU as the activation function. The 8 × 1 feature space is reconstructed into 16 × 1, 32 × 1, and 64 × 1 shapes by the decoder, and the reconstructed signal of 100 × 1 is finally output by the output layer. In the decoding process, each decoder will jointly learn the output characteristics of the corresponding encoder while receiving the upper layer input. The network can better reconstruct the original signal, accelerate the convergence speed, and alleviate the gradient loss problem [18].

### 2.3. Training Process

In the training process, the training samples $X=\left\{x_{1},x_{2},\ldots ,x_{n}\right\}$ only contain normal signals, which are input into the ResNet-AE neural network model. The residual structure extracts the sample features through the convolution and LSTM layers. The main features of the training samples are mapped by rules φ to the feature space $Y=\left\{y_{1},y_{2},\ldots ,y_{n}\right\}$. As shown in Formula (1), $W^{\left(1\right)}$ and $b^{\left(1\right)}$ represent the weight and offset from the input layer to the coding layer [19].
(1)$$y\left(i\right)=\varphi \left(W^{\left(1\right)}x\left(i\right)+b^{\left(1\right)}\right)$$

At the same time, we can learn that the residual error of the sample is,
(2)F(x)=Y(x)−x

The original learning feature is F(x)+x. When the residual is 0, the residual structure only performs identity mapping, and the network performance will not decline. However, in the actual process, the residual will not be 0, making the residual structure learn new features based on the input features, so it performs better. The residual element is shown in Formulas (3) and (4), where $n_{l}$ and $n_{l+1}$, respectively, represent the input and output of the l residual unit, and each residual unit contains a multilayer structure. F is the residual function, representing the learned residual; and $h\left(n_{l}\right)=n_{l}$ represents the identity mapping; and f is the ReLU activation function [20].
(3)$$m_{l}=h\left(n_{l}\right)+F\left(n_{l},W_{l}\right)$$
(4)$$n_{l+1}=f\left(m_{l}\right)$$

Based on the above formula, the learning characteristics from shallow l to deep L are
(5)$$n_{L}=n_{l}+\overset{L-1}{\underset{i=l}{\sum }}F\left(n_{i},W_{i}\right)$$

The reconstructed features are realized by multiple total connection layers and deconvolution layers through the inverse mapping rules ψ. Restore and reconstruct the feature space data and reconstruct the space $\hat{X}=\left\{{\hat{x}}_{1},{\hat{x}}_{2},\ldots ,{\hat{x}}_{n}\right\}$ consistent with the sample space dimension. As shown in Formula (6), $W^{\left(2\right)}$ and $b^{\left(2\right)}$ are the weights and offsets from the coding layer to the output layer.
(6)$$\hat{x}\left(i\right)=\varphi \left(W^{\left(2\right)}y\left(i\right)+b^{\left(2\right)}\right)$$

ResNet-AE updates the model parameters according to the loss function. Learning here aims to minimize the distance between the reconstructed output data and the incoming and outgoing data. The loss function can be defined as
(7)$$F\left(\theta \right)=arg_{\theta }\underset{j}{\sum }∥x_{j}-{\hat{x}}_{j}{∥}_{2}^{2}$$

### 2.4. Anomaly Decision

At the end of the training phase, the weights and offsets of the network model are determined. At this time, each layer node of the neural network can be regarded as the expression of the input signal in different feature spaces. Since the training process uses normal signal data, the model can only extract the features of normal signals. The mapping of anomalies in the feature space is distorted, resulting in redundancy and loss [21]. The characteristic component of the anomaly is distorted after the convolution network, which cannot be mapped to the feature space, let alone reconstructed by the autoencoder. The reconstruction effect of the network on anomaly is inferior. Therefore, whether there is an anomaly in the input signal subsequence can be judged by the reconstruction effect of the network model on the data to be measured, and the reconstruction effect here can be evaluated by the distance between the reconstructed and the original data. Suppose that the input sample data are $X=\left\{x_{1},x_{2},\ldots ,x_{n}\right\}$, and the reconstruction output of the neural network is $Y=\left\{y_{1},y_{2},\ldots ,y_{n}\right\}$. The reconstruction error can be expressed as the mean square error
(8)$$error=\frac{1}{n}\overset{n}{\underset{i=1}{\sum }}{\left(y_{i}-x_{i}\right)}^{2}$$

The reconstruction error of the neural network model for abnormal and normal signals is quite different. The error for the anomaly is generally large, and the error for the normal signal should be close to 0 [22]. Based on this property, the threshold T can be reasonably set as the decision threshold. The decision process is as follows
(9)$$result\{\begin{array}{c} error<T,\;0\\ error>T,\;1 \end{array}$$

In the above formula, “0” indicates that the sample is normal, and “1” indicates an anomaly.

## 3. Experiment

This experiment is divided into four stages: data preprocessing, model training, model testing, and model evaluation. The hardware and software environment of the experiment are shown in Table 1.

### 3.1. Data Preprocessing

The main task of the data preprocessing stage is to convert the original signal data sequence into a dataset that the neural network can receive. By setting the threshold values of the rising and falling edges of the pulses, each pulse signal is extracted into time sequence subsequence samples. Then, these are divided into the training set, verification set, and test set according to 6:2:2. The training set and verification set only contain normal signals. The test set is a mixture of normal and anomalies. Anomalies include partial loss, mutation, or strong noise interference of signals. At the same time, two other groups of experiments are set. In the first group, 90 dB, 60 dB, and 30 dB noise are added to all the data, respectively, to carry out the same independent repeated experiment. The second group of experiments is the radar signal data generated by five different radars, and the experiments are repeated independently.

### 3.2. Model Training

Because there are few hyperparameters, we use grid search to select hyperparameters. First, define the traversal interval Batchsize = {16, 32, 64, 128}, Learning rate = {0.0001, 0.001, 0.01, 0.1}, and Loss function = {MSE}, Optimizer = {Adma}, and then calculate the cost function of all hyperparametric combinations on the validation set to obtain the optimal hyperparametric set in the interval. Epoch is determined by observing the convergence of the loss function. The final hyperparameters are shown in Table 2:

The encoder learns the training data to obtain the feature space. Then, the decoder reconstructs the feature into the source data. The loss function is calculated and optimized iteratively until the model error reaches the expectation and the final weight model is saved. The algorithm flow of model training is shown in Figure 3:

After data preprocessing operations such as subsequence division and normalization, the training samples are input into the ResNet-AE network for feature extraction, and a feature space of 8 × 1 is obtained. After the reconstruction of the decoder, the feature vector is restored to the dimension of the original data to obtain the reconstructed sequence signal. The comparison between the reconstructed signal and the input signal is shown in Figure 4. In the figure, blue is the original signal, and red is the reconstructed signal. The figure shows the reconstructed signal when the epoch is 0, 50, 100, and 150, respectively. When the epoch is 0, because the model’s weight at the initial training stage is a random value, the reconstructed signal greatly differs from the original signal. With the continuous iteration and optimization of parameters, when the epoch is 50, the reconstruction error of the network reaches a low level, and the original signal can be reconstructed well. Until the end of the training, the reconstruction error is unchanged, and the reconstruction effect tends to be stable. It can be seen from the figure that for normal signals, the ResNet-AE model can be well reconstructed, and the reconstructed signals are consistent with the original signals.

Figure 5 shows the changes in loss functions during the training of several common models. Among them, the loss functions of the four models converge rapidly. When the epoch reaches 150, the ResNet-AE model tends to be flat and stable, and the loss functions of the AE and CNN-AE models still have a downward trend. Although the loss functions of VAE converge rapidly, there are small fluctuations and fluctuations. At the same time, the loss functions of stable models are large, indicating that the network reconstruction effect is poor. It can be seen that the ResNet-AE model has certain advantages in the reconstruction of normal signals.

### 3.3. Model Testing

In the test phase, first, read the weight model saved in the training phase, input the test set data after data preprocessing into the model to obtain the error value, and perform K-Means clustering on the error value [23]. All error values are divided into two categories according to the size of the value. The small category is determined as a normal signal, the large category is determined as an anomaly, and the adaptive threshold of abnormal judgment is obtained. The algorithm flow of the model test is shown in Figure 6:

The relevant data are visually analyzed to show the results of model anomaly detection. Figure 7 shows the anomaly and its reconstructed signal. It can be seen from the figure that the anomaly will be distorted after the reconstruction of ResNet-AE [24], and the original signal cannot be restored. The reconstructed signal has a large error from the original signal, so the normal and anomalies can be effectively distinguished.

By calculating the mean square error between the signal subsequence in the test set and its reconstructed signal, the abnormal score of the signal is obtained, and a K-Means classifier with two categories is constructed. The random initial clustering center is used to cluster the abnormal score. After iteration, the clustering center of the normal signal score and the anomaly score is finally obtained. The mean value of the two clustering centers is the threshold value of abnormal judgment. Figure 8 shows the anomaly detection scores of each subsequence. The red horizontal line is the adaptive threshold, the signal above the threshold is the anomaly, and the signal below the threshold is the normal signal. The adaptive threshold obtained through cluster analysis can significantly distinguish normal and anomalies.

### 3.4. Model Evaluation

In this experiment, the ResNet-AE anomaly detection model will be evaluated by five indicators: *Accuracy*, *Precision*, *Recall*, *F*1 score, and AUC [25].
(10)$$Accuracy=\frac{TP+TN}{TN+TP+FN+FP}$$
(11)$$Precision=\frac{TP}{TP+FP}$$
(12)$$Recall=\frac{TP}{TP+FN}$$
(13)$$F1=\frac{2\times Precision*\times \;Recall}{Precision+Recall}$$

*TN*: the number of signals which are predicted as normal signals and actually normal signals, that is, the prediction of the algorithm is correct;

*TP*: the number of signals which are predicted as anomalies but actually normal signals, that is, the algorithm predicts correctly;

*FN*: the number of signals which are predicted as normal signals but actually anomalies, that is, the algorithm predicts incorrectly;

*FP*: the number of signals which are predicted as anomalies and actually anomalies, that is, the algorithm predicts incorrectly.

## 4. Results and Analysis

We compared the ResNet-AE model with the common AE and GAN models to verify its performance. At the same time, to verify its generalization ability, experiments were carried out on different signal-to-noise ratios and equipment signal data. Random noise is added to the original signal, and the signal-to-noise ratio is 90 dB, 60 dB, and 30 dB, respectively. The influence of the signal-to-noise ratio of the radar signal on the abnormal detection results is explored; Five groups of signals generated by different radar emitters are selected for experiments, and the anomaly detection results of different radar signals are recorded. It is verified that the ResNet-AE model still has good anomaly detection ability on unfamiliar time-series signals.

### 4.1. Comparison of Common Models

The reconstructed signal output of common anomaly detection models is shown in Figure 9. For normal signals, the network can reconstruct the original signal. The reconstructed signal is similar to the original signal in shape, and the reconstruction ability of each network is different. It can be seen from the figure that the linear AE network has a poor reconstruction effect. Many positions and shapes between the reconstructed and original signals cannot coincide. The network can not be reconstructed for anomalies, which are very different from the original signal. Compare the detection performance of the ResNet-AE model and several common models, as shown in Figure 10. Compared with other traditional AE models [26,27,28], ResNet-AE has greater advantages in various indicators. Compared with several GAN models [29,30], ResNet-AE also has a certain improvement in detection *Accuracy* and *F*1 value, reaching a higher level. From the anomaly detection results of each network, it can be seen that the network with CNN, LSTM, and ResNet structures have higher *Accuracy* than the linear network. Therefore, the features of time-series radar signals can be better extracted by using these structures. In addition, considering the practical application, it is expected to find all anomalies as much as possible. Because the cost of judging anomalies as normal is higher, we should focus on considering *Recall*, reducing missed detection first, and allowing a certain amount of false alarms. The *Recall*s of each network in the experiment are maintained high. It indicates a high recognition rate for anomalies. The possibility of missed detection is low, and the *Precision* is quite different. The network will judge some normal signals as abnormal, and there are false alarms.

The complexity of the model is evaluated by calculating the FLOPs and Params of the commonly used anomaly detection models. Figure 11 shows the logarithm of the FLOPs and Params of each model. The FLOPs and Params of the GAN-based model are 2–3 orders of magnitude higher than those of the AE-based model. It indicates that the complexity of the GAN-based model is much higher than that of the AE-based model; The relevant numerical difference of the model based on AE is within one order of magnitude, and the complexity is equivalent. Therefore, the ResNet-AE model not only ensures high detection capability but also has a more concise model structure.

### 4.2. Influence of SNR

This experiment explores the influence of abnormal detection results on the noise of radar signals. Gaussian white noise is added to the original signal, and the SNRs are 90 dB, 60 dB, and 30 dB, respectively. Repeat the above experimental process to obtain the experimental results, and compare them with the experimental results of the original signal. Figure 12 shows the signal reconstruction effect. As the signal-to-noise ratio decreases, the difference between the reconstructed and original signals becomes larger. When the signal-to-noise ratio is 90 dB and 60 dB, the original signal can still be reconstructed, and the error change is not obvious; When the signal-to-noise ratio reaches 30 dB, the reconstructed signal is completely distorted. Figure 13 shows various evaluation indicators under different signal-to-noise ratios. The *Accuracy*, *Precision*, and *F*1 values generally show a downward trend with the reduction of SNR but remain at a high level. When the signal-to-noise ratio is reduced to 30 dB, the *Accuracy* and *Precision* decline significantly, indicating that at the 30 dB signal-to-noise ratio level, noise greatly interferes with the abnormal detection of signals. Due to the increase in noise, it is difficult to extract the characteristics of signals, thus affecting the error of reconstructed signals. As a result, some normal signals are difficult to reconstruct and are judged as abnormal, which has little impact on detecting anomalies, so the change in *Recall* is relatively stable. The model performs well under different signal-to-noise ratios and can adapt to data sequences with low signal-to-noise ratios.

### 4.3. Generality Analysis

In this experiment, five groups of signals generated by different radar radiation sources are selected for experiments, and the anomaly detection results of different radar signals are recorded. As shown in Figure 14, the training set is generated by Equipment 1, and its reconstruction effect is significantly better than that of other equipment. The signals generated by other equipment are unfamiliar signals, of which the reconstruction effect of Signal 2 is ideal. The reconstruction effect of other signals is general. The difference between the reconstructed and original signals mainly lies in the rising or falling edge. The reconstruction effect is better in the middle of the signal. Figure 15 shows the various evaluation indicators of different radar signals. Since Equipment 1 is the equipment for generating training set signals, all indicators in the five groups of signals are the highest. Although the *Accuracy*, *Precision*, and *F*1 values of other signals are slightly lower than Signal 1, they are all above 0.84, reaching a high level. At the same time, the *Recall*s of the five groups of signals are all above 0.94, retaining the network’s ability to distinguish anomalies. It also maintains the characteristics of a low missed detection rate for unfamiliar signals, and the values of various indicators are stable, which shows that the model has good anomaly detection ability and strong universality.

## 5. Conclusions

This paper proposes a radar signal anomaly detection model based on ResNet-AE. Based on the traditional autoencoder, the convolution neural network is used to extract features and learn the potential distribution law of data. We use LSTM to learn the time dependence of data and use residual structure to alleviate the missing gradient problem, improve the use efficiency of the depth network, and use mean square deviation to make anomaly judgments. The adaptive threshold obtained by clustering is used as the benchmark of anomaly decision, which can distinguish the normal signal from the anomaly and improve the *Accuracy* of anomaly decision. Compared with several commonly used AE and GAN models for anomaly detection, the model has certain advantages in *Accuracy*, *Precision*, *Recall*, *F*1 value, AUC value, and other evaluation indicators. At the same time, the model has good performance in different signal-to-noise ratios and different radar equipment, and has certain universality.

However, the model still has some limitations. It can only detect whether it is an anomaly and cannot classify the types of anomalies in a more detailed way. Therefore, identifying the types of anomalies, that is, whether the anomalies are caused by equipment failures, natural environmental factors, or human interference, is the future research direction [31,32]. It can provide more effective support and help analyze potential battlefield threats. Moreover, our method is mainly applicable to the FMCW radar system. When the center frequency, bandwidth, pulse width, and other parameters change, the feasibility of the method will be verified one by one in the subsequent work so as to improve the generalization ability of the network to apply to more types of radar signals.

## Figures and Tables

**Figure 1 sensors-22-06249-f001:**
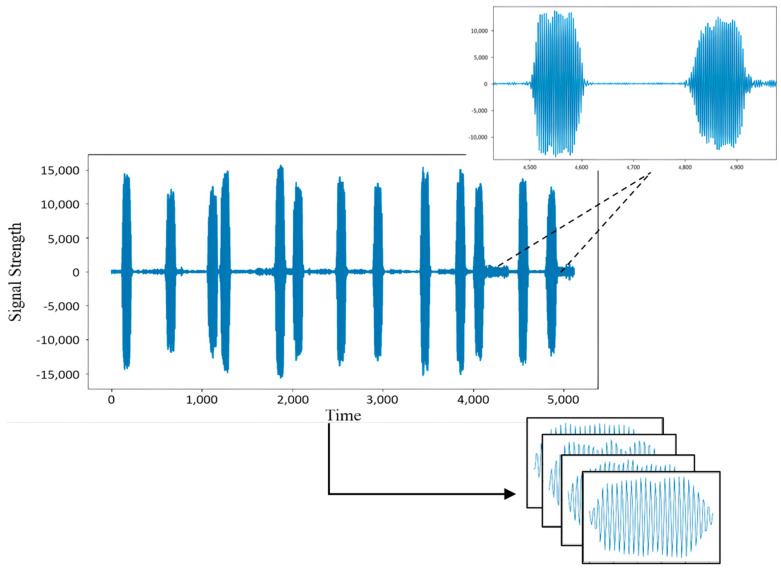
Radar signal dataset.

**Figure 2 sensors-22-06249-f002:**
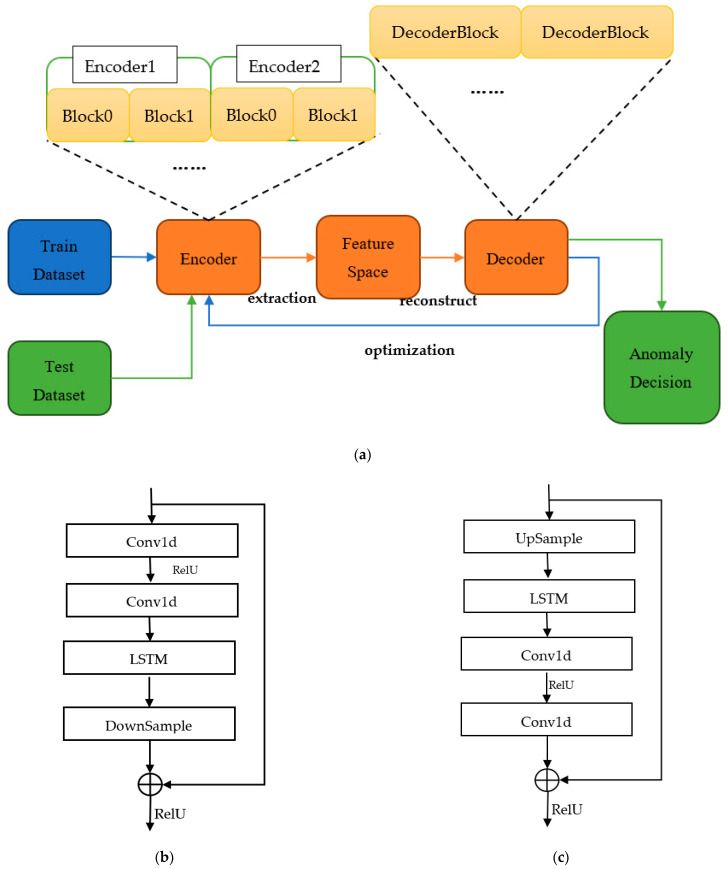
ResNet-AE network structure: (**a**) Overall structure; (**b**) Encoder residual structure; (**c**) Decoder residual structure.

**Figure 3 sensors-22-06249-f003:**
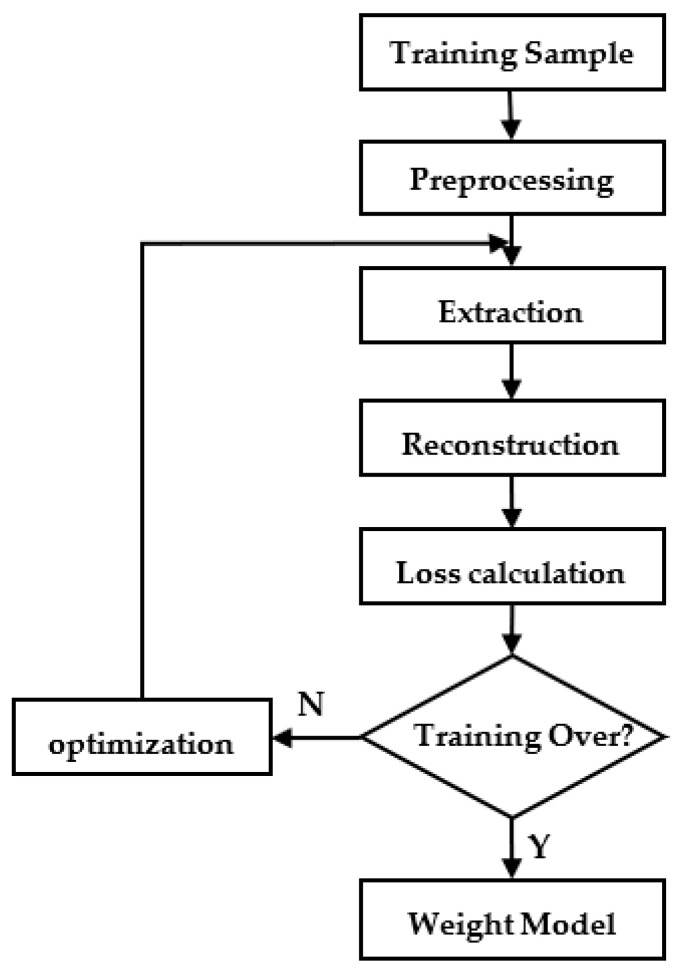
Model training algorithm flow.

**Figure 4 sensors-22-06249-f004:**
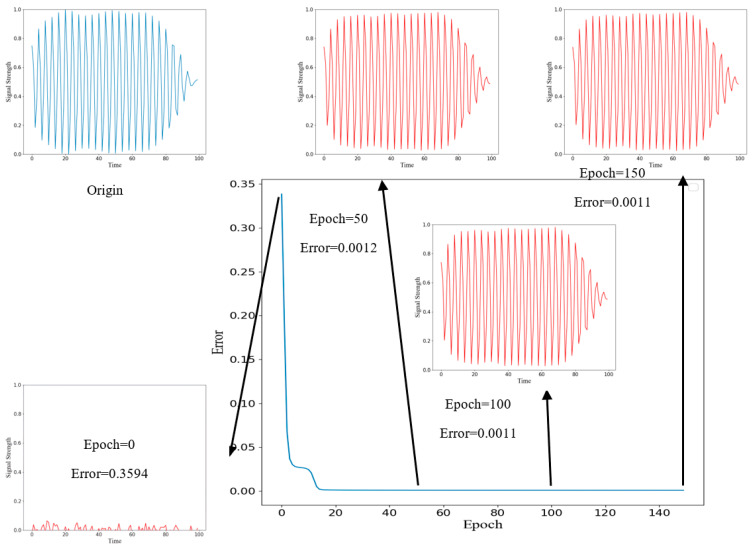
Reconstructed signal during training.

**Figure 5 sensors-22-06249-f005:**
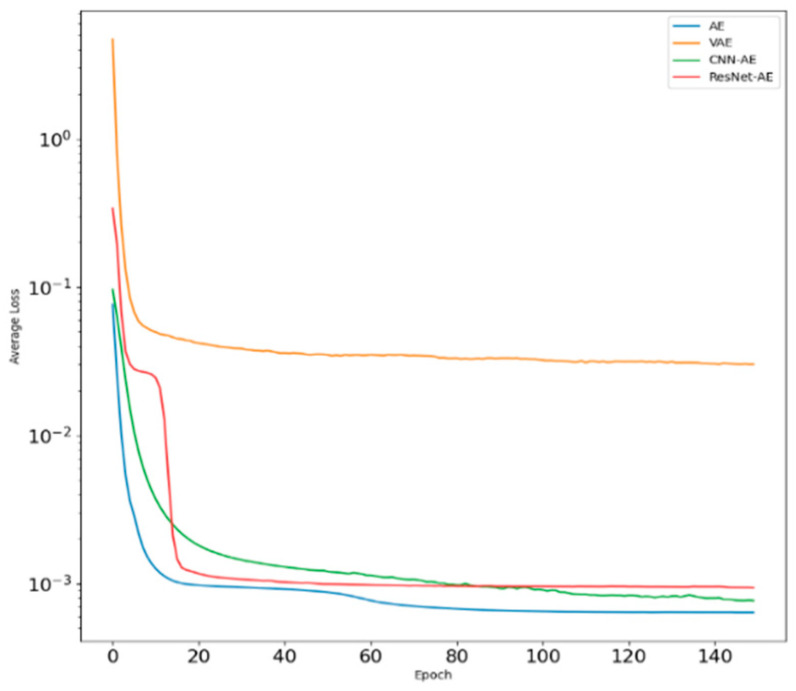
Variation of training loss of several models.

**Figure 6 sensors-22-06249-f006:**
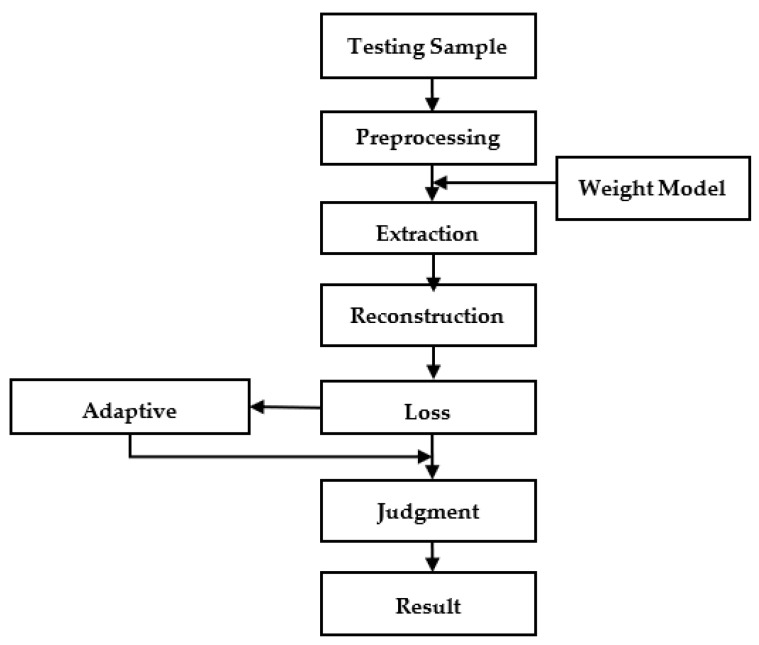
Model testing algorithm flow.

**Figure 7 sensors-22-06249-f007:**
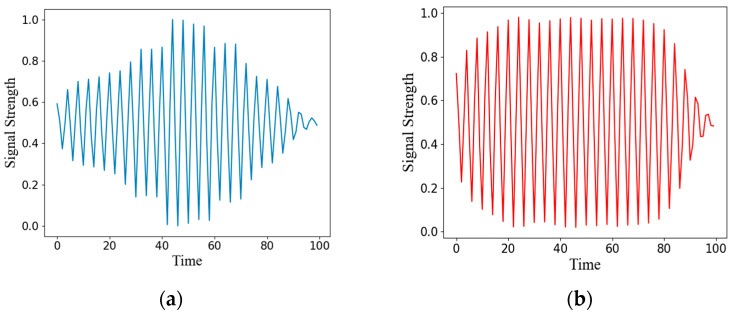
Anomaly and reconstructed signal: (**a**) Original signal of an anomaly; (**b**) Reconstruction signal of anomaly.

**Figure 8 sensors-22-06249-f008:**
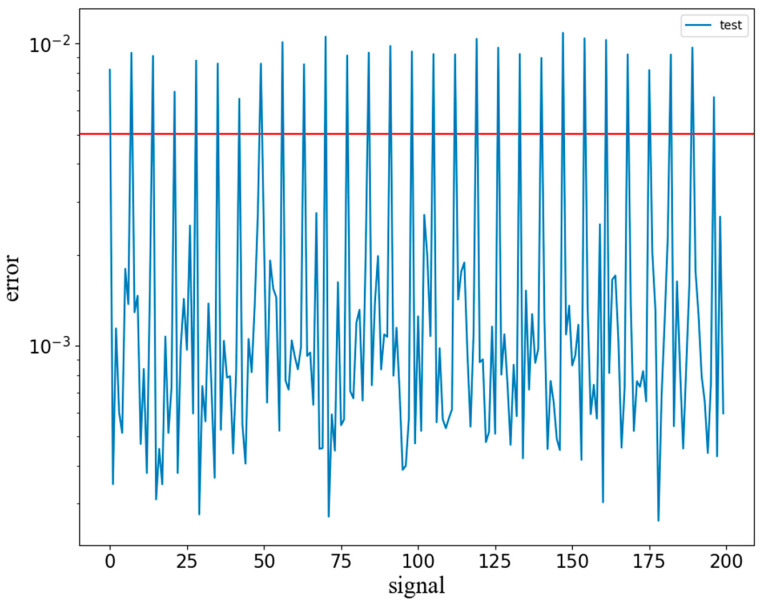
Abnormal judgment.

**Figure 9 sensors-22-06249-f009:**
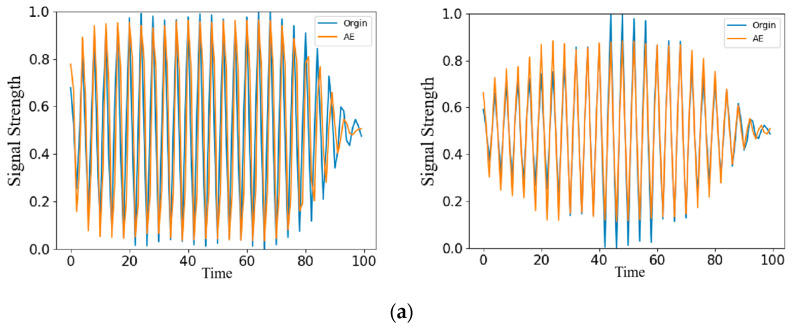
Reconstruction signal of common anomaly detection models: (**a**) Abnormal detection results of AE; (**b**) Abnormal detection results of VAE; (**c**) Abnormal detection results of CNN-AE; (**d**) Abnormal detection results of LSTM-AE; (**e**) Abnormal detection results of ResNet-AE.

**Figure 10 sensors-22-06249-f010:**
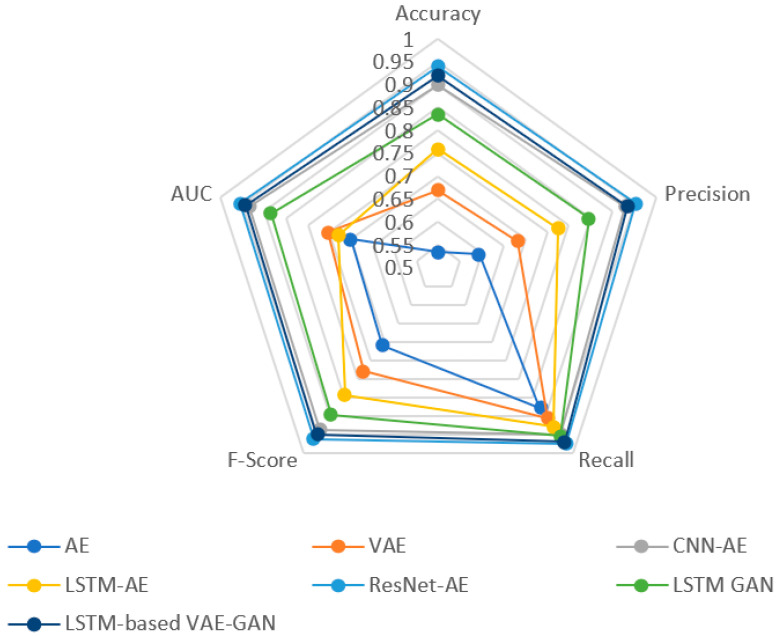
Result evaluation of common anomaly detection models.

**Figure 11 sensors-22-06249-f011:**
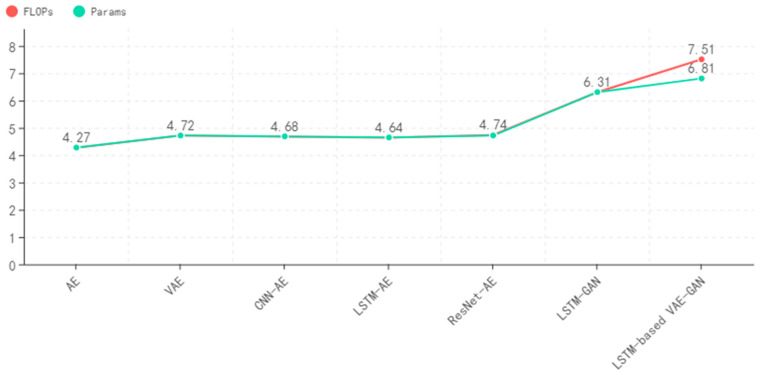
Complexity analysis of common anomaly detection models.

**Figure 12 sensors-22-06249-f012:**
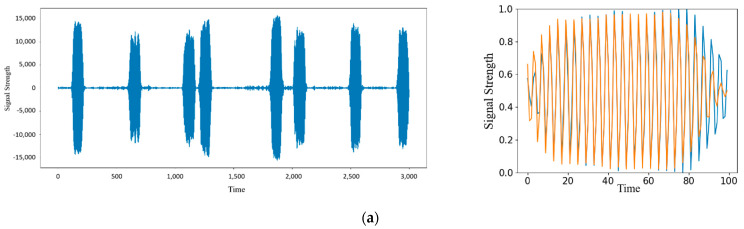
Reconstructed signals with different SNR: (**a**) Original signal and abnormal detection results; (**b**) Signal with SNR = 90 dB and its anomaly detection results; (**c**) Signal with SNR = 60 dB and its anomaly detection results; (**d**) Signal with SNR = 30 dB and its anomaly detection results.

**Figure 13 sensors-22-06249-f013:**
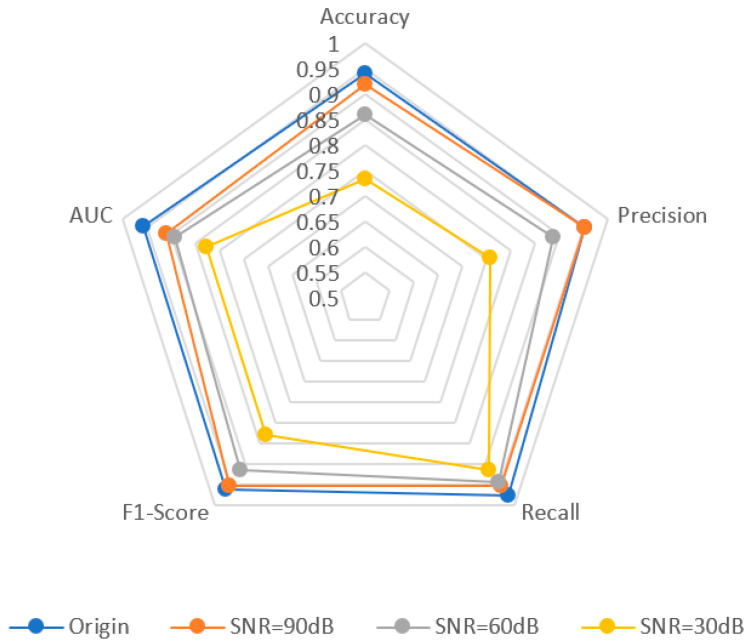
Evaluation of anomaly detection results under different signal-to-noise ratios.

**Figure 14 sensors-22-06249-f014:**
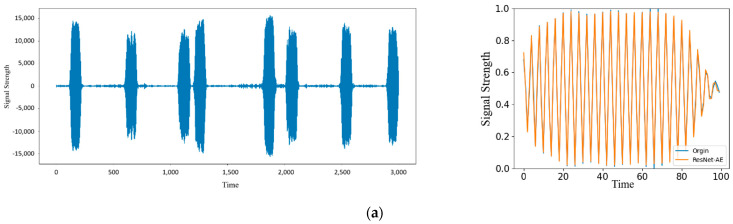
Reconstructed signals of different radar: (**a**) Signal 1 and its abnormal detection results; (**b**) Signal 2 and its abnormal detection results; (**c**) Signal 3 and its abnormal detection results; (**d**) Signal 4 and its abnormal detection results; (**e**) Signal 5 and its abnormal detection results.

**Figure 15 sensors-22-06249-f015:**
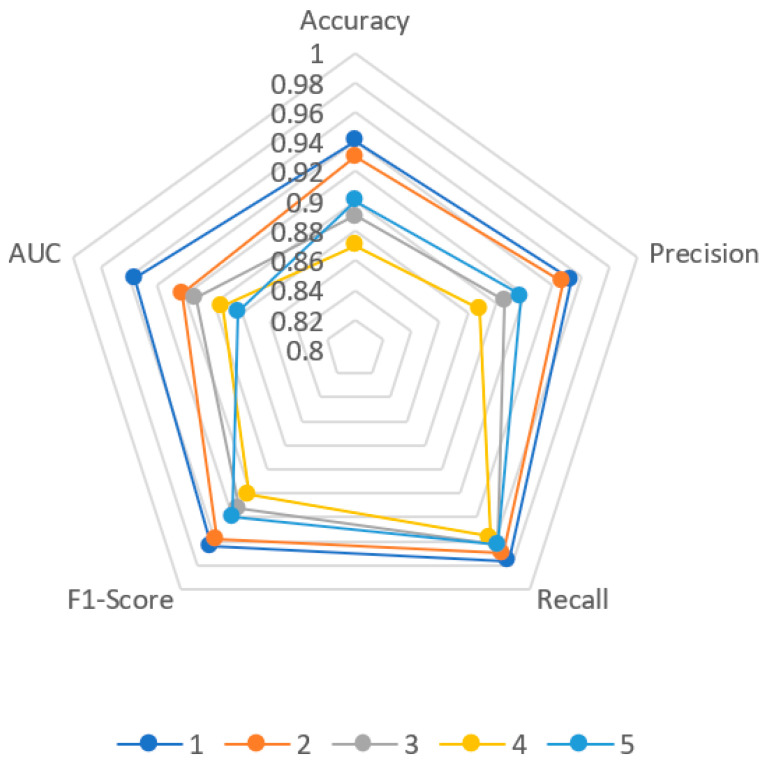
Evaluation of abnormal detection results of different radar.

**Table 1 sensors-22-06249-t001:** Configuration of hardware and software.

Hardware or Software	Technical Parameter	Hardware or Software
Operation System	Windows 10 Home Chinese	Operation System
CPU	Intel Core i5-10300H	CPU
GPU	NVIDIA Geforce RTX 2060	GPU
Memory	32 G	Memory
Python	Python 3.8.5	Python
Pytorch	Pytorch 1.6.0	Pytorch

**Table 2 sensors-22-06249-t002:** Hyperparameter of ResNet-AE.

Hyperparameter	
Batch size	64
Epochs	150
Learning rate	0.001
Loss function	MSE
Optimizer	Adam

## Data Availability

Not applicable.

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
