# Peer review of "ResNet-AE for Radar Signal Anomaly Detection"

_sensors, 2022, doi:10.3390/s22166249_

Round 1

Reviewer 1 Report

1. The goal of the thesis is to detect anomaly of radar signals, but there is too little description of the radar system. Among radar systems such as CW, FMCW, and pulse-Doppler, it is necessary to mention which system is the main target for application. In addition, various evaluations are needed on whether the proposed method is effectively applied even when the center frequency, bandwidth, and pulse width are changed in each radar system.

2. The authors use the terms abnormal and anomal interchangeably, but in fact, the two have different strict meanings. The authors must correct the terms appropriately.

3. The procedure for determining the hyperparameters of ResNet-AE used by the authors is not presented. In the radar dataset used by the authors, performance may be further improved if a different hyperparameter set is used.

4. The resolution of the figures in the paper is too low to read the axis information. Authors should redraw pictures with resolution problems. In addition, the English expression in the manuscript is too poor, so it is essential to have it corrected by a native speaker.

It is thought that review will be possible again after these points are corrected.

Reviewer 2 Report

The following suggestions aimed at improve the quality of the manuscript must be taken into account by the authors:

1) Please better specify in the introduction section the innovation introduced in this work because in this form they are not sufficiently clear. 

2) an accurate analysis of the computational time and resources required to apply the proposed method should be reported. The proposed method if i understand correctly improve only of 4% with respect to the other methodology so you must justify the advantages of using your new methodology.

3)Figures2 should be better commented as well as the Resnet AE anomaly detection model. Please enclose further descriptions.

4) The reference section requires to be improved it turn out to be too much limited. 

Round 2

Reviewer 1 Report

The authors well responded to my comments.

Reviewer 2 Report

The revised version of the manuscript has been improved and now I think that it is acceptable for publication. Please check the body of the manuscript and remove minor grammar and typos errors.